# Third-Order Edge Statistics: Contour Continuation, Curvature, and Cortical Connections

**Matthew Lawlor**
Applied Mathematics
Yale University
New Haven, CT 06520
matthew.lawlor@yale.edu

**Steven W. Zucker**
Computer Science
Yale University
New Haven, CT 06520
zucker@cs.yale.edu

## Abstract

Association field models have attempted to explain human contour grouping performance, and to explain the mean frequency of long-range horizontal connections across cortical columns in V1. However, association fields only depend on the pairwise statistics of edges in natural scenes. We develop a spectral test of the sufficiency of pairwise statistics and show there is significant higher order structure. An analysis using a probabilistic spectral embedding reveals curvature-dependent components.

## 1 Introduction

Natural scene statistics have been used to explain a variety of neural structures. Driven by the hypothesis that early layers of visual processing seek an efficient representation of natural scene structure, decorrelating or reducing statistical dependencies between subunits provides insight into retinal ganglion cells [17], cortical simple cells [13, 2], and the firing patterns of larger ensembles [18]. In contrast to these statistical models, the role of neural circuits can be characterized functionally [3, 14] by positing roles such as denoising, structure enhancement, and geometric computations. Such models are based on evidence of excitatory connections among co-linear and co-circular neurons [5], as well as the presence of co-linearity and co-circularity of edges in natural images [8], [7]. The fact that statistical relationships have a geometric structure is not surprising: To the extent that the natural world consists largely of piecewise smooth objects, the boundaries of those objects should consist of piecewise smooth curves.

Common patterns between excitatory neural connections, co-occurrence statistics, and the geometry of smooth surfaces suggests that the functional and statistical approaches can be linked. Statistical questions about edge distributions in natural images have differential geometric analogues, such as the distribution of intrinsic derivatives in natural objects. From this perspective, previous studies of natural image statistics have primarily examined "second-order" differential properties of curves; i.e., the average change in orientation along curve segments in natural scenes. The pairwise statistics suggest that curves tend toward co-linearity, in that the (average) change in orientation is small. Similarly, for long-range horizontal connections, cells with similar orientation preference tend to be connected to each other.

Is this all there is? From a geometric perspective, do curves in natural scenes exhibit continuity in curvatures, or just in orientation? Are edge statistics well characterized at second-order? Does the same hold for textures?

To answer these questions one needs to examine higher-order statistics of natural scenes, but this is extremely difficult computationally. One possibility is to design specialized patterns, such as intensity textures [16], but it is difficult to generalize such results into visual cortex. We make use of natural invariances in image statistics to develop a novel spectral technique based on preserving

a probabilistic distance. This distance characterizes what is beyond association field models (discussed next) to reveal the "third-order" structure in edge distributions. It has different implications for contours and textures and, more generally, for learning.

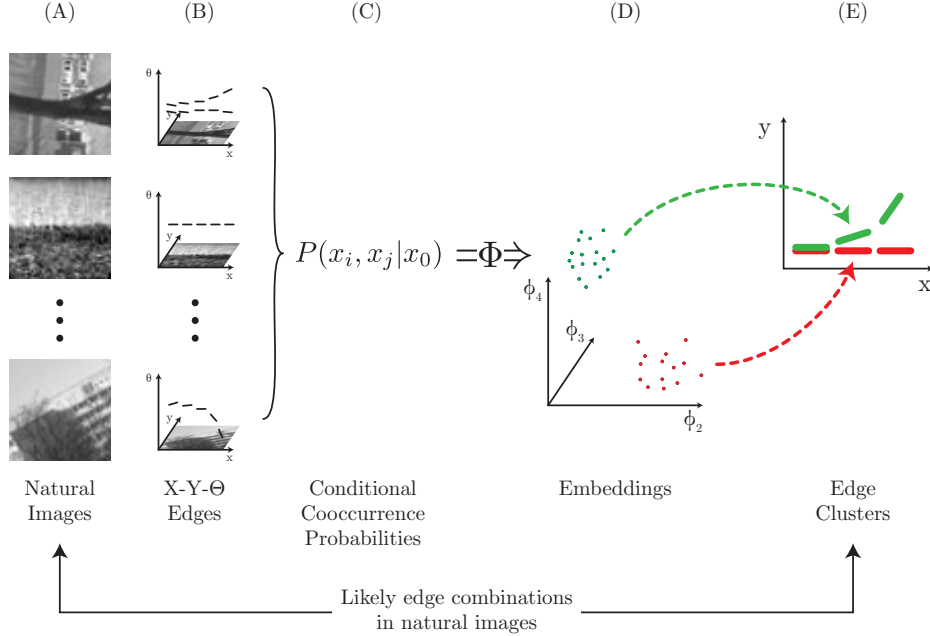

Figure 1: Outline of paper: We construct edge maps from a large database of natural images, and estimate the distribution of edge triplets. To visualize this distribution, we construct an embedding which reveals likely triplets of edges. Clusters in this embedded space consist of curved lines

## 2 Edge Co-occurrence Statistics

Edge co-occurrence probabilities are well studied [1, 8, 6, 11]. Following them, we use random variables indicating edges at given locations and orientations. More precisely, an edge at position, orientation $r_i = (x_i, y_i, \theta_i)$, denoted $X_{r_i}$, is a $\{0, 1\}$ valued random variable. Co-occurrence statistics examine various aspects of pairwise marginal distributions, which we denote by $P(X_{r_i}, X_{r_j})$.

The image formation process endows scene statistics with a natural translation invariance. If the camera were allowed to rotate randomly about the focal axis, natural scene statistics would also have a rotational invariance. For computational convenience, we enforce this rotational invariance by randomly rotating our images. Thus,

$$P(X_{r_1}, ..., X_{r_n}) = P(X_{T(r_1)}, ..., X_{T(r_n)})$$

where $T$ is a roto-translation.

We can then estimate joint distributions of nearby edges by looking at patches of edges centered at a (position, orientation) location $r_n$ and rotating the patch into a canonical orientation and position that we denote $r_0$. Let $T(r_n) = r_0$. Then

$$P(X_{r_1}, ..., X_{r_n}) = P(X_{T(r_1)}, ..., X_{r_0})$$

Several examples of statistics derived from the distribution of $P(X_{r_i}, X_{r_0})$ are shown in Fig. 2. These are pairwise statistics of oriented edges in natural images. The most important visible feature of these pairwise statistics is that of *good continuation*: Conditioned on the presence of an edge at the center, edges of similar orientation and horizontally aligned with the edge at the center have high probability. Note that all of the above implicitly or explicit enforced rotation invariance, either by

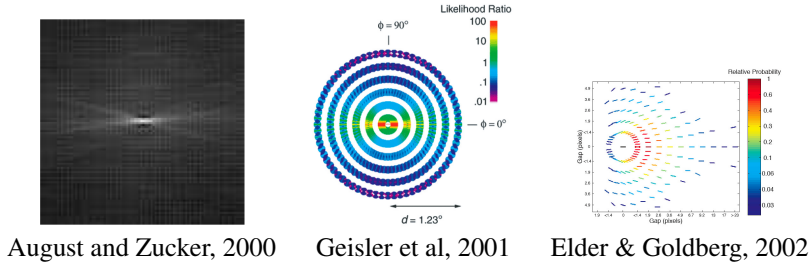

| August and Zucker, 2000 | Geisler et al, 2001 | Elder & Goldberg, 2002 |

Figure 2: Association fields derive from image co-occurrence statistics. Here we show three attempts to characterize them. Different authors consider probabilities or likelihoods; Elder further conditions on boundaries. We simply interpret them as illustrating the probability (likelihood) of an edge near a horizontal edge at the center position.

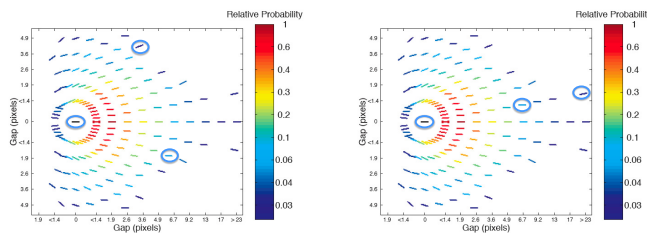

Figure 3: Two approximately equally likely triples of edges under the pairwise independence assumption of Elder et. al. Conditional independence is one of several possible pairwise distributional assumptions. Intuitively, however, the second triple is much more likely. We examine third-order statistics to demonstrate that this is in fact the case.

only examining relative orientation with respect to a reference orientation or by explicit rotation of the images.

It is critical to estimate the degree to which these pairwise statistics characterize the full joint distribution of edges (Fig. 3). Many models for neural firing patterns imply relatively low order joint statistics. For example, spin-glass models [15] imply pairwise statistics are sufficient, while Markov random fields have an order determined by the size of neighborhood cliques.

## 3   Contingency Table Analysis

To test whether the joint distribution of edges can be well described by pairwise statistics, we performed a contingency table analysis of edge triples at two different threshold levels from images in the van Hataran database. We computed estimated joint distributions for each triple of edges in an $11 \times 11 \times 8$ patch, not constructed to have an edge at the center. Using a $\chi^2$ test, we computed the probability that each edge triple distribution could occur under hypothesis $H_0 : \{$No three way interaction$\}$. This is a test of the hypothesis that

$$\log P(X_{r_i}, X_{r_j}, X_{r_k}) = f(X_{r_i}, X_{r_j}) + g(X_{r_j}, X_{r_k}) + h(X_{r_i}, X_{r_k})$$

for each triple $(X_{r_i}, X_{r_j}, X_{r_k})$, and includes the cases of independent edges, conditionally independent edges, and other pairwise interactions. *For almost all triples, this probability was extremely small.* (The few edge triples for which the null hypothesis cannot be rejected consisted of edges that were spaced very far apart, which are far more likely to be nearly statistically independent of one another.)

| $n = 150705016$ | threshold $= .05$ | threshold $= .1$ |
|---|---|---|
| percentage of triples where $p_{H_0} > .05$ | 0.0082% | 0.0067% |

## 4  Counting Triple Probabilities

We chose a random sampling of black and white images from the van Hataren image dataset[10]. They were randomly rotated and then filtered using oriented Gabor filters covering 8 angles from $[0, \pi)$. Each Gabor has a carrier period of 1.5 pixels per radian and an envelope standard deviation of 5 pixels. The filters were convolved in near quadrature pairs, squared and summed.

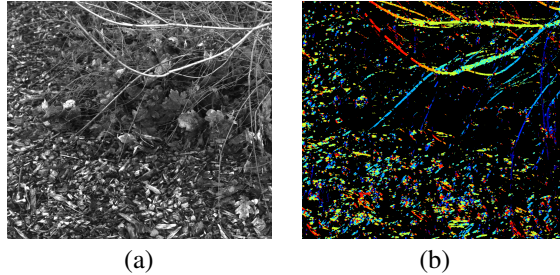

(a)                    (b)

Figure 4: Example image (a) and edges (b) for statistical analysis. Note: color corresponds to orientation

To restrict analysis to the statistics of curves, we applied local non-maxima suppression across orientation columns in a direction normal to the given orientation. This threshold is a heuristic attempt to exclude non-isolated curves due to dense textures. We note that previous studies in pairwise edge statistics have used similar heuristics or hand labeling of edges to eliminate textures. The resulting edge maps were subsampled to eliminate statistical dependence due to overlapping filters.

Thresholding the edge map yields $X : U \to \{0, 1\}$, where $U \subset \mathbb{R}^2 \times \mathbb{S}$ is a discretization of $\mathbb{R}^2 \times \mathbb{S}$. We treat $X$ as a function or a binary vector as convenient. We randomly select $21 \times 21 \times 8$ image patches with an oriented edge at the center, and denote these characteristic patches by $V_i$

Since edges are significantly less frequent than their absence, we focus on (positive) edge co-occurrence statistics. For simplicity, we denote $P(X_{r_i} = 1, X_{r_j} = 1, X_{r_k} = 1)$ by $E[X_{r_i} X_{r_j} X_{r_k}]$. In addition, we will denote the event $X_{r_i} = 1$ by $Y_{r_i}$. (A small orientation anisotropy has been reported in natural scenes (e.g., [9]), but does not appear in our data because we effectively averaged over orientations by randomly rotating the images.)

We compute the matrix $M^+$ where

$$M_{ij}^+ = E[X_{r_i} X_{r_j} | Y_{r_0}]$$
$$\sim \frac{1}{n} \sum_{i=1}^{n} V_i V_i^T$$

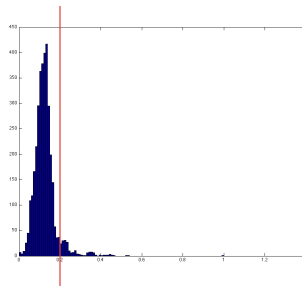

Figure 5: Histogram of edge probabilities. The threshold to include an edge in $M^+$ is $p > 0.2$, and is marked in red.

where $V_i$ is a (vectorized) random patch of edges centered around an edge with orientation $\theta_i = 0$. In addition, we only compute pairwise probabilities for edges of high marginal probability (Fig. 5)

# 5 Visualizing Triples of Edges

By analogy with the pairwise analysis above, we seek to find those edge triples that frequently co-occur. But this is significantly more challenging. For pairwise statistics, one simply fixes an edge to lie in the center and "colors" the other edge by the joint probability of the co-occurring pair (Fig. 2). No such plot exists for triples of edges. Even after conditioning, there are over 12 million edge triples to consider.

Our trick: *Embed edges in a low dimensional space such that the* distance *between the edges represents the relative likelihood of co-occurrence.* We shall do this in a manner such that *distance in Embedded Space $\sim$ Relative Probability.*

As before, let $X_{r_i}$ be a binary random variable, where $X_{r_i} = 1$ means there is an edge at location $r_i = (x_i, y_i, \theta_i)$. We define a distance between edges

$$D_+^2(r_i, r_j) = E[X_{r_i}^2|Y_{r_0}] - 2E[X_{r_i}X_{r_j}|Y_{r_0}] + E[X_{r_j}^2|Y_{r_0}]$$
$$= M_{ii}^+ - 2M_{ij}^+ + M_{jj}^+$$

The first and the last terms represent pairwise co-occurrence probabilities; i.e., these are the association field. The middle term represents the interaction between $X_{r_i}$ and $X_{r_j}$ conditioned on the presence of $X_0$. Thus this distance is zero if the edges always co-occur in images, given the horizontal edge at the origin, and is large if the pair of edges frequently occur with the horizontal edge but rarely together. (The relevance to learning is discussed below.)

We will now show how, for natural images, edges can be placed in a low dimensional space where the distance in that space will be proportional to this probabilistic distance.

# 6 Dimensionality Reduction via Spectral Theorem

We exploit the fact that $M^+$ is symmetric and introduce the spectral expansion

$$M^+ = \sum_{l=1}^{n} \lambda_l \phi_l(i)\phi_l(j)$$

where $\phi_l$ is an eigenvector of $M^+$.

Define the *spectral embedding* $\Phi : \begin{pmatrix} x_i \\ y_i \\ \theta_i \end{pmatrix} \rightarrow \mathbb{R}^n$

$$\Phi(r_i) = \{\sqrt{\lambda_1}\phi_1(i), \sqrt{\lambda_2}\phi_2(i), ..., \sqrt{\lambda_n}\phi_n(i)\} \tag{1}$$

The Euclidean distance between embedded points is then

$$\|\Phi(r_i) - \Phi(r_j)\|^2 = \langle \Phi(r_i), \Phi(r_i)\rangle - 2\langle \Phi(r_i), \Phi(r_j)\rangle + \langle \Phi(r_j), \Phi(r_j)\rangle$$
$$= M_{ii}^+ - 2M_{ij}^+ + M_{jj}^+$$
$$= D_+^2(r_i, r_j)$$

$\Phi$ maps edges to points in an embedded space where squared distance is equal to relative probability.

The usefulness of this embedding comes from the fact that the spectrum of $M^+$ decays rapidly (Fig. 6). Therefore we truncate $\Phi$, including only dimensions with high eigenvalues. This gives a dramatic reduction in dimensionality, and allows us to visualize the relationship between triples of edges (Fig. 7). In particular, a cluster, say, $C$, of edges in embedding space all have high probability of co-occurring, and the diameter of the cluster

$$d = \max_{i,j \in C} D^2(r_i, r_j)$$

bounds the conditional co-occurrence probability of all edges in the cluster.

$$E[X_{r_i}, X_{r_j}|Y_{r_0}] \geq \frac{2p - d}{2}$$

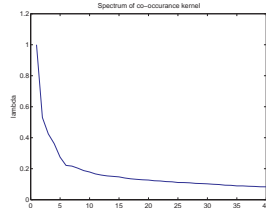

Figure 6: Spectrum of $M^+$. Other spectra are similar. Note rapid decay of the spectrum indicating the diffusion distance is well captured by embedding using only the first few eigenfunctions.

Spectral embedding colored by embedding coordinates

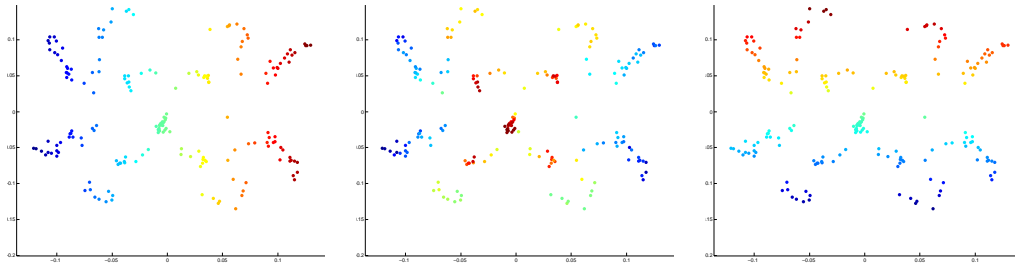

Edge map colored by embedding coordinates

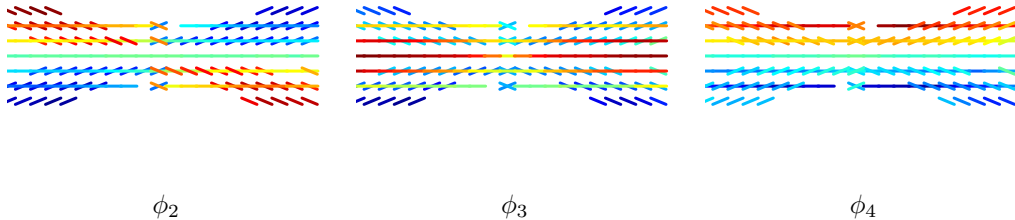

$\phi_2$ $\qquad\qquad\qquad\qquad\qquad$ $\phi_3$ $\qquad\qquad\qquad\qquad\qquad$ $\phi_4$

Figure 7: Display of third-order edge structure showing how oriented edges are related to their spectral embeddings. (top) Spectral embeddings. Note clusters of co-occurring edges. (bottom) Edge distributions. The eigenvectors of $M^+$ are used to color both the edges and the embedding. The color in each figure can be interpreted as a coordinate given by one of the $\phi$ vectors. Edges that share colors (coordinates) in all dimensions $(\phi_2, \phi_3, \phi_4)$ are close in probabilistic distance, which implies they have a *high probability of co-occurring* along with the edge in the center. Compare with Fig. 2 where red edges all have high probability of occurring with the center, but no information is known about their co-occurrence probability.

where $p = \min_i E(X_{r_i}|Y_{r_0})$. For our embeddings $p > .2$ see Fig. 5.

To highlight information not contained in the association field, we normalized our probability matrix by its row sums, and removed all low-probability edges. Embedding the mapping from $\mathbb{R}^2 \times \mathbb{S} \to \mathbb{R}^m$ reveals the cocircular structure of edge triples in the image data (Fig. 7). The colors along each column correspond, so similar colors map to nearby points along the dimension corresponding to the row. Under this dimensionality reduction, each small cluster in diffusion space corresponds to half of a cocircular field. In effect, the coloring by $\phi_2$ shows good continuation in orientation (with our crude quantization) while the coloring by $\phi_4$ shows co-circular connections. In effect, then, the

association field is the union of co-circular connections, which also follows from marginalizing the third-order structure away. We used 40,000 ($21 \times 21 \times 8$) patches.

Shown in Fig. 7 are low dimensional projections of the diffusion map and their corresponding colorings in $\mathbb{R}^2 \times \mathbb{S}$. To provide a neural interpretation of these results, let each point in $\mathbb{R}^2 \times \mathbb{S}$ represent a neuron with a receptive field centered at the point $(x, y)$ with preferred orientation $\theta$. Each cluster then signifies those neurons that have a high probability of co-firing given that the central neuron fires, so clusters in diffusion coordinates should be "wired" together by the Hebbian postulate. Such curvature-based facilitation can explain the non-monotonic variance in excitatory long-range horizontal connections in V1 [3, 4]. It may also have implications for the receptive fields of V2 neurons. As clusters of co-circular V1 cells are correlated in their firing, it may be efficient to represent them with a single cell with excitatory feedforward connections. This predicts that efficient coding models that take high order interactions into account should exhibit cells tuned to curved boundaries.

# 7  Implications for Inhibition and Texture

Our approach also has implications beyond excitatory connections for boundary facilitation. We repeated our conditional spectral embedding, but now conditioned on the *absence* of an edge at the center (Fig. 8). This could provide a model for inhibition, as clusters of edges in this embedding are likely to co-occur conditioned on the absence of an edge at the center. We find that the embedding has no natural clustering. Compared to excitatory connections, this suggests that inhibition is relatively unstructured, and agrees with many neurobiological studies.

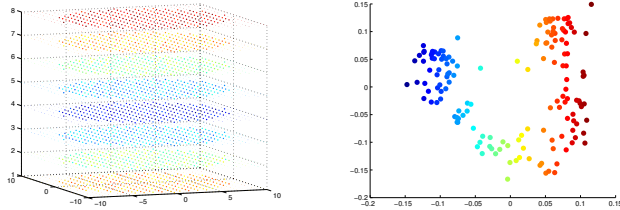

Figure 8: Embeddings conditioned on the absence of an edge at the center location. Note how less structured it is, compared to the positive embeddings. As such it could serve as a model for inhibitory connections, which span many orientations.

Finally, we repeated this third-order analysis (but without local non-maxima suppression) on a structured model for isotropic textures on 3D surfaces and again found a curvature dependency (Fig. 9). Every 3-D surface has a pair of associated dense texture flows in the image plane that correspond to the slant and tilt directions of the surface. For isotropic textures, the slant direction corresponds to the most likely orientation signaled by oriented filters.

As this is a representation of a dense vector field, it is more difficult to interpret than the edge map. We therefore applied k-means clustering in the embedded space and segmented the resulting vector field. The resulting clusters show two-sided continuation of the texture flow with a fixed tangential curvature (Fig. 10).

In summary, then, we have developed a method for revealing third-order orientation structure by spectral methods. It is based on a diffusion metric that makes third-order terms explicit, and yields a Euclidean distance measure by which edges can be clustered. Given that long-range horizontal connections are consistent with these clusters, how biological learning algorithms converge to them remains an open question. Given that research in computational neuroscience is turning to third-order [12] and specialized interactions, this question now becomes more pressing.

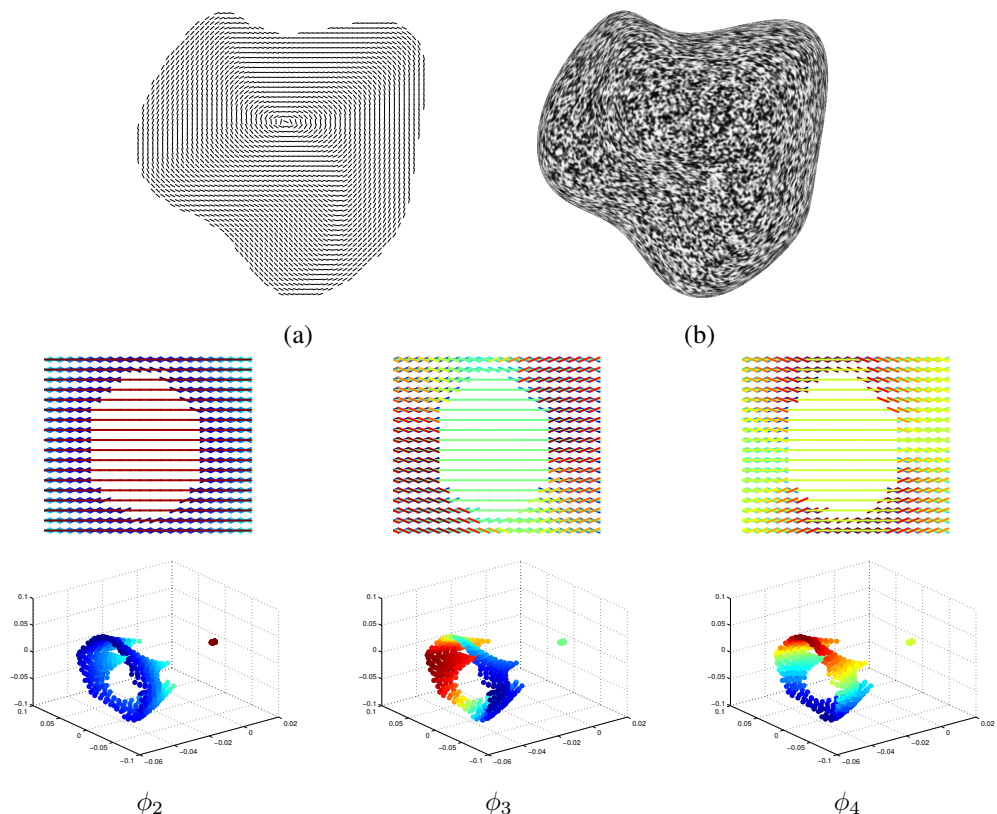

$\phi_2$          $\phi_3$          $\phi_4$

Figure 9: (top) Oriented textures provide information about surface shape. (bottom) As before, we looked at the conditional co-occurrence matrices of edge orientations over a series of randomly generated shapes. Slant orientations and embedding colored by each eigenvector. The edge map is thresholded to contain only orientations of high probability. The resulting embedding $\phi(v_i)$ of those orientations is shown below. The eigenvectors of $M^+$ are used to color both the orientations and the embedding. Clusters of orientations in this embedding have a *high probability of co-occurring* along with the edge in the center.

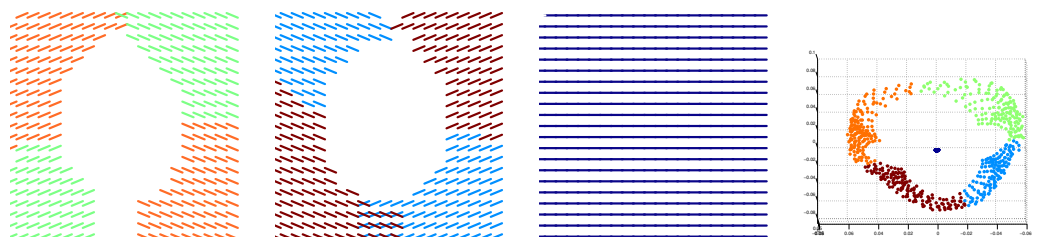

Figure 10: Clustering of dense texture flows. Color corresponds to the cluster index. Clusters were separated into different figures so as to minimize the $x, y$ overlap of the orientations. Embedding on the right is identical to the embeddings above, but viewed along the $\phi_3, \phi_4$ axes.

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
