[Reviews · NeurIPS 2013]

Submitted by Assigned_Reviewer_1

This paper attempts to model and visualize the third order statistics of edge occurrence in natural images. Because directly doing this is hard, the authors only compute the conditional distribution given a central edge with orientation zero. Then, in order to present the distribution the resulting matrix is projected on its first eigenvectors which are the subspace in which most of the variance is. The authors show that the points close in that subspace are also close in probability space. Results are calculated for non-maxima suppressed filter response images (see below) and for textures.

Quality:
The idea and motivations for this work are clear, but I have several reservations of other aspects of this work. First and foremost, applying non-maxima suppression for the filter response images is, I think, fundamentally flawed - if one needs to study the nature of edge co-occurrence in natural images I would expect one to leave the images as "natural" as possible. While this experiment is repeated in Section 5 of the paper without non-maxima suppression, a large part of the paper's message relies on these flawed experiments and this takes away from the main message.
Additionally, the spectral decomposition is presented as a novelty, but this is just a common dimensionality reduction procedure, nothing to be so detailed about.

Clarity:
A weak point of the paper - notation is quite misleading, there are several different notations for the same element which can be confusing (Ei,Ej,Ek --> i,j,k) etc. The introductory text could also use clearer explanations of the basic ideas - there is an extra page the authors use to make this clearer.

Originality:
Seems like an original work all in all, though the spectral dimensionality reduction part is really overstressed.

Significance:
This is basic important research question which could be relevant to a large part of the vision community, though I am not sure if this work actually answers it.
Summary: A nice, basic, work which is interesting and significant, but suffers from some weaknesses in the experimentation and presentation.

Submitted by Assigned_Reviewer_4

Authors analyze the statistics of trios of edges in natural images. This calculation explodes if you try to do it by just counting all the possibile entries in the table of 12 million possible entries. So they approximate the calculation by a spectral method, visualizing the resulting statistics with a 2-d embedding (found from keeping only strong spectral terms). The authors find that there is 3rd order structure in edges statistics. They then claim this will have implications in neural organization, based on the Hebbian postulate that cells that fire together wire together.

Quality: I think this is good work, but not pushed far enough. Given the quantitative analysis, I wanted more quantitative conclusions and predications. Those are all very hand wavy in the "implications" section.

clarity: It's hard to squeeze this into a conference paper and these results could work better in a journal format. Many numerical values (required for reproducibility) were missing, eg, line 263 "we...removed all low-probability edges" what criterion did you use? How much of the variance is explained by the 2-d embedding you ended up with?

Originality: This seems like an obvious extension of 2nd order edge statistics, so I would be surprised it hasn't been studied before, but I can't locate a reference on it and it may be original. The embedding approach to approximating these statistics was not obvious to me.

Significance: If the narrative-style results were presented in a more quantitative way, it could give the paper more significance.
Summary: Interesting paper, perhaps important, Needs more details given in order to be reproducible by others. More quantitative results would make this a stronger paper.

Submitted by Assigned_Reviewer_5

( Required, Visible To Authors During Feedback and After Decision Notification )

Synopsis.
This paper examines third-order co-occurrence statistics of edges in natural images. The main finding is that triples of edges co-occur in ways consistent with smooth flows, elongated along the tangent direction.

Better statistical models of the geometry of natural image structure are crucial to understanding biological vision as well as making progress in computer vision, and the authors are correct in their claim that to date most models have focused on 2nd-order relationships, particularly the joint distribution of pairs of edges. Thus the investigation of higher-order statistics is of great interest.

Quality.
Overall the methods seem sound.

Clarity.
The paper is understandable in broad strokes, but there are many parts that are unclear, and in general the paper is a bit sloppy and hurried in presentation. Here are some examples:

• Line 74. I don’t think this equation is what the authors intend.
• Line 106. What data were used for this analysis? Later in Section 4, the authors indicate that they use the van Hataren dataset, but it’s not clear whether this is also used here.
• Line 205 What is the K matrix? Looks to me like the phis are just the eigenvectors of the P(i,j|0) matrix.
• Line 232 Do you mean Fig 4
• Line 234 Do you mean Fig 5?
• Fig 5. This figure is very unclear. What are the dimensions in the top panels? The first two eigenvectors?
• Lines 264. From the text I cannot understand the color coding scheme. “The colors along each column correspond, so similar colors map to nearby points along the dimension corresponding to the row” – Columns of what? Rows of what? Clearly the colors are somehow dependent on phi_2, phi_3 and phi_4 in the left, middle and right columns. I assume then, that the color is determined by the value of the projection of each point on these eigenvectors?
• Line 249. The authors suggest that the projection on phi_2 reveals colinearity and the projection on phi_4 indicates cocircularity. What then does the projection on phi_3 represent?
• …”each small cluster in diffusion space corresponds to half of a cocircular field”. What is a cocircular field? Where is the proof of this statement?
• Line 274 “Shown above..” above where?
• Line 280 Why complex cells and not simple cells? The assumed circuit model should be specified.
• “Such curvature-based facilitation can explain the non-monotonic variance in excitatory long-range horizontal connections in V1”. This will be completely impenetrable except for a very small group of cognoscenti.
• Line 288. We find that the embedding is only one dimensional, with the significant eigenvector a monotonic function…” This is an example of methodological sloppiness. What are the criteria for determining dimensionality and significance?
• Line 298? What does the figure on the left represent, and what am I supposed to learn from it?
• Line 312, “the distribution of these directions in 3D shape are an excellent proxy for measuring the response of orientation filters to dense textured objects”. Why do I want a proxy for measuring orientation filter responses? Aren’t these the proximal measurements I can actually measure, to make inference about the distal 3D shape variables I cannot directly measure?
• Line 346. Why are their two clusters (one cylinder, one spot)?

Originality.
There isn’t that much work on higher-order geometry statistics, so in this sense the paper is fairly novel.

Significance.
Although the effort here is worthwhile, I do think significance is lessened a bit because a) the results are not particularly surprising or counter to current thinking, b) there is not a clear advance in terms of hypothesis testing, improved quantitative model or algorithm and c) the paper is unclear in places. On the other hand, perhaps the novel spectral embedding method could in the future be used to achieve a more clear contribution.

Also, when judging significance, it’s important to make clear that most prior work on natural edge statistics does not really make the claim that 2nd-order statistics are sufficient models for joint edge statistics in natural images.

For example, Geisler et al and Elder & Goldberg were not measuring co-occurrence statistics over images. Geisler was measuring co-occurrence statistics over object boundaries, and Elder & Goldberg were measuring sequential statistics along object boundaries. Thus the Elder & Goldberg data, for example, provide the statistics required for a first-order Markov model of contours, not a Markov model of the 2D edge (association) field. Thus in Fig 2, the conditional distribution for contours is being misapplied to the problem of modeling the edge field.

To make this clearer, note that generally the E&G model would predict the existence of higher-order co-occurrence statistics over the edge field. For example, suppose this model is used to generate a number of contours sparsely distributed over the image. The edge elements belonging to the same contour would generate strong third-order co-occurrences explainable by second-order statistics along the contour but NOT explainable by second order statistics in the 2D association field.

This does not mean that the present effort to understand these third-order co-occurrences in the edge field is not worthwhile, but it’s important to put it in the right context. In particular, given this presentation it is still not clear whether a model like the E&G model might be sufficient to explain the 2D edge field. (Although we know this is unlikely: Ren & Malik (2002) have in fact shown that contours are not 1st-order Markov – no surprise.)

X. F. Ren and J. Malik. A probabilistic multi-scale model for contour completion based on image statistics. In Lecture Notes in Computer Science, Proc. ECCV, volume 2350, pages 312–327, Berlin, 2002. Springer-Verlag.
Summary: Interesting problem, and spectral embedding method may be of value. However the results are unsurprising, not clearly presented, and no clear advance in hypothesis testing, model or algorithm is achieved. On the other hand, perhaps the novel spectral embedding method could in the future be used to achieve a more clear contribution.
Author Feedback

Author rebuttal: We thank the reviewers for their (mostly) positive comments on our paper, and apologize
for the sloppiness in its preparation. The notation, figures/captions, etc. will be improved
should we be given the opportunity.

We chose a discursive style thinking that we could carry
the reader along a path from dimensionality reduction to learning in neurobiology, because this
rests on the distance measure that arises in Sec 5. (We believe this distance is an important
part of the paper, and one that is not usually developed.) Our goal was not to develop new algorithms for
curve detection. Moreover, following learning in biology, we did not wish to use labeled data for
contours.

To expand: Normally, to apply dimensionality
reduction arguments one might represent image (edge) patches as points in 4K-dimensions
(21 x 21 pixels x 10 orientations) and reduce this: the result would
yield a distance measure over edge patches.

What we are doing is different. The motivation is, simply put: if neurons represent edges, and if those that 'fire together wire together,'
then 'frequently co-occuring edge pairs' becomes a surrogate for 'fire together.' Our distance measure
was designed to represent this by summarizing the edge statistics of patches into a probability matrix;
we then find a low rank approximation to it. This yields the distance over conditioned pairs, and it relates
to the association field and second-order statistics.

Stated differently: if one builds a data matrix whose columns are the vectorized edge characteristic function (so there
are 4K rows and the number-of-patches columns, then (after normalization) we are doing dimensionality reduction
on the rows.

The beauty of our construction is that it extends, by the conditioning argument, to (some) third-order statistics.

By looking at pairwise probabilities conditioned on a central edge we were able to find some of the geometric
and continuity properties as clusters of edges in embedded coordinates (Fig 5 - 8). It is these clusters that
(we believe) visual systems learn. (There was no space in this paper for our learning algorithm.) This includes
co-linearity (Fig 5, \phi_2) co-circularity (\phi_4), plus some measure of straight vs curved (\phi_3). The half-fields
suggest the Markov property discussed by Reviewer 3.

Figure 5 has two rows of sub-figures: the top are the embedded edge points and the bottom is a illustration of them
in image, orientation terms. Clusters in the top row correspond to those edges that should be 'wired' together. The coordinates are
\phi_2, \phi_4. The caption, coordinates, and labeling of this figure will be improved substantially.

Reviewer 1 asks about non-maxima suppression. Because we are working toward biological learning of connections
but have no hand-labeled curves, we needed a technique to separate the dense edges that arise from textures from
the 1D distributions of edges along curves. Non-maxima suppression does this in a simple fashion.
We will add a figure with embeddings without the non-maxima suppression.

We are gratified that all reviewers agreed third-order edge statistics are important, and look forward
to tightening our presentation. While Elder and Goldberg might have predicted the existence of higher-order
edge statistics, we know of no other study that finds and characterizes them. We will
place the distance measure more prominently in the Introduction.